# Early Soviet satellite magnetic field measurements from 1964 and 1970

Roman Krasnoperov[1], Dmitry Peregoudov[1], Renata Lukianova[1], Anatoly Soloviev[1,2] and Boris Dzeboev[1]

[1]Geophysical Center of the Russian Academy of Sciences, Moscow, 119296, Russia
[2]Schmidt Institute of Physics of the Earth of the Russian Academy of Sciences, Moscow, 123242, Russia

*Correspondence to*: Dmitry Peregoudov (d.peregoudov@gcras.ru)

**Abstract.** We present the collection of magnetic field absolute measurements performed by early Soviet magnetic satellite missions Kosmos-49 (1964) and Kosmos-321 (1970). A total of 17300 measured values are available for Kosmos-49 mission, covering homogeneously 75% of the Earth's surface between 49° north and south latitude. About 5000 measured values are available for Kosmos-321 mission, covering homogeneously 94% of the Earth's surface between 71° north and south latitude. The data are available at PANGAEA (Krasnoperov et al., 2020, https://doi.org/10.1594/PANGAEA.907927).

## 1 Introduction

Since 1954, the Soviet Union was an active participant in the preparation of the International Geophysical Year (IGY), organized in 1957–1958. Right from the beginning, the program of this unprecedent international scientific event included the launch of an artificial satellite with a payload for conducting geophysical experiments on the Earth's orbit. The first Soviet missiles of the 1950-s allowed to perform solely suborbital flights. And only by 1957, the R-7 ballistic missile, capable of launching an object into a circular orbit, had been developed. The first Soviet spacecraft, widely known as Sputnik-1, was launched on 04.10.1957. Its payload consisted only of two radio transmitters that continuously emitted signals on two frequencies. Receiving signals on different frequencies allowed the analysis of the radiowave propagation in the ionosphere. The second artificial satellite Sputnik-2 that was launched on 3.11.1957 had a more sophisticated scientific payload including instruments for registration of cosmic rays' and solar radiation parameters and an air-tight container for a biological experiment with a dog. In accordance with the program of the IGY the third Soviet spacecraft – Sputnik-3 – was launched on 15.05.1958. Its payload weighed 968 kg and included equipment for 12 scientific experiments. Amongst other instruments was a unique self-orienting fluxgate magnetometer that performed first orbital measurements of the Earth's magnetic field total intensity (Petrukovich, 2009; Skuridin, 1975).

# 1 Satellite missions for magnetic field measurements

## 1.1 Kosmos-49

Since 1960-s the Soviet design office OKB-586 (currently – M.K. Yangel Yuzhnoye Design Office, Ukraine) initiated the development of a new experimental series of small-scale spacecraft, designated as DS (Dnepropetrovsk Satellite). The DS-series satellites had a unified platform with a standard on-board control system suite. In 1962 the launch of the first DS spacecraft started the Soviet space research program "Kosmos" under which a large number, over 2500, of various satellite missions have been performed. The payload of the spacecraft varied and, depending on the purpose, provided a wide range of scientific experiments: astronomical, astrophysical, geophysical, ionospheric, atmospheric, meteorite, radiation, etc. (Konyukhov, 2000).

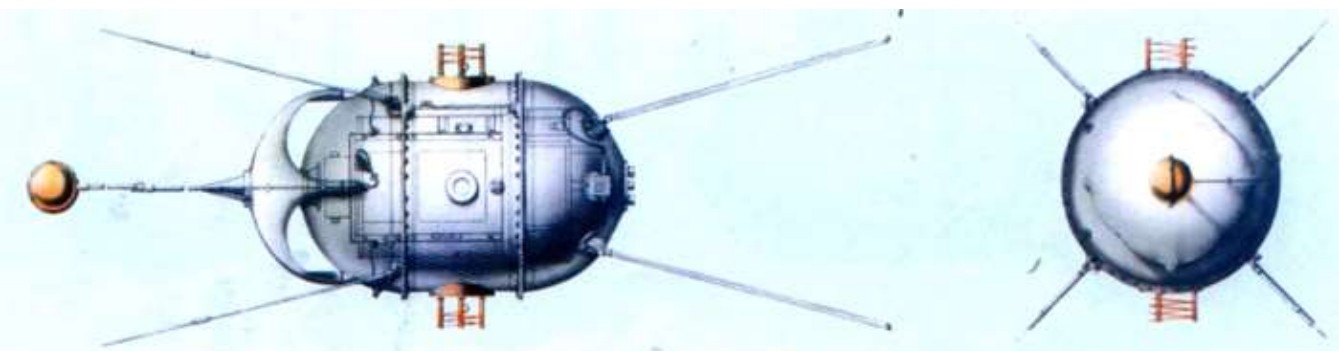

**Figure 1: External view of the DS-MG type spacecraft (Kosmos-26 and Kosmos-49), 1964 (Konyukhov, 2000).**

The satellite mission aimed at obtaining the direct data on the spatial distribution of the Earth's magnetic field in the course of the global geomagnetic survey consisted of two spacecraft of DS-MG (MG – MaGnetic) type that were designated Kosmos-26 and Kosmos-49 and launched on 18.03.1964 and 24.10.1964 respectively from Kapustin Yar cosmodrome (Astrakhan region). The design and payload of both spacecraft was identical and included a set of two absolute proton precession magnetometers developed by the Institute of Terrestrial Magnetism, Ionosphere and Radio Wave Propagation of the Academy of Sciences of the USSR (IZMIRAN) and manufactured by the Special Design Bureau "Geophysics" of the Ministry of Geodesy of the USSR. The instrumental accuracy of the magnetometers was estimated at the level of 2–3 nT (Cain, 1971; Dolginov et al.,1966). Considering all the external effects and internal sources of errors the characteristic of the total error with which Kosmos-49 surveyed the magnetic field was estimated at the level of 25–30 nT (Benkova and Dolginov, 1971). The sensors of two devices were oriented at the right angle to each other. The devices were switched on and off alternately for 32 seconds intervals. Approximately 62% of the measured values were recorded and are available in the catalogue, published by IZMIRAN (Dolginov et al., 1967). The external view of the DS-MG type spacecraft is given in Fig. 1. Kosmos-49 had cylindrical body with the diameter of 1.2 m and length of 1.8 m. Magnetometers were mounted 3.3 m

from the center of the satellite. The magnetic effects of its body were compensated to accuracy of 2 nT by the system of permanent magnets.

The mission of Kosmos-26 and Kosmos-49 confirmed the possibility of using Earth's magnetic field data for determination of spacecraft orientation. The obtained geomagnetic data justified the evidence of propagation of magnetic anomalies, associated with the structure and tectonics of the Earth's crust, to the heights of low-orbiting satellites.

## 1.2 Kosmos-321

The success of the first mission with experimental DS-series spacecraft also confirmed the feasibility of remote sensing methods for solving a great variety of scientific problems. The Academy of Sciences of the USSR developed technical specifications for a new series of unified spacecraft that included three standard platforms: DS-U1 (with chemical energy source), DS-U2 (with solar panels without self-orientation), DS-U3 (with solar panels and self-orientation). The on-board control and supply systems' segment had standardized construction and was independent from the specific mission payload segment. This allowed the organization of the serial production of spacecraft and their components reducing financial and time expenditure. The follow-on mission aimed at global geomagnetic survey of the Earth also consisted of two spacecraft of a new DS-U2-MG type designated as Kosmos-321 and Kosmos-356 and launched on 20.01.1970 and 10.08.1970 respectively from Plesetsk cosmodrome (Arkhangelsk region). The design of both spacecraft was identical: cylindrical body with the diameter of 0.8 m and length of 1.46 m, consisting of a cylindrical shell and two hemispherical bottoms. The spacecraft had three separate compartments for scientific equipment, support systems and power supply systems.

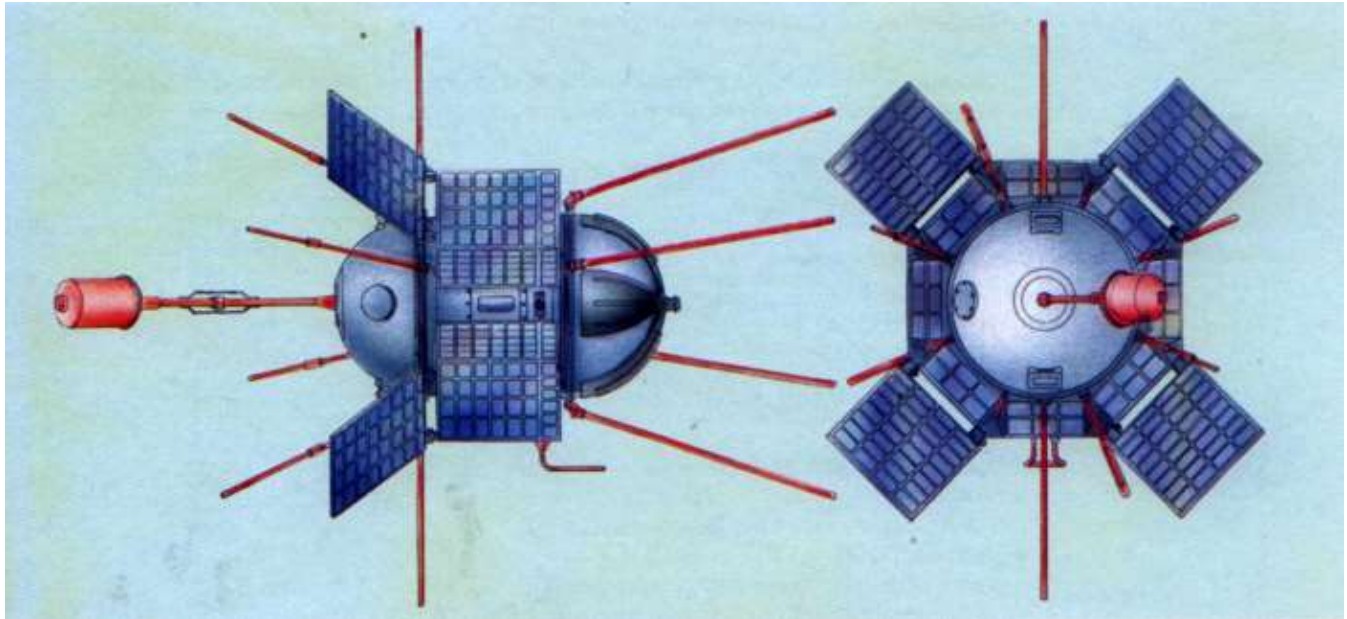

**Figure 2: External view of the DS-U2-MG type spacecraft (Kosmos-321 and Kosmos-356), 1970 (Konyukhov, 2000).**

The mission payload included a self-generated quantum magnetometer with optical pumping in cesium vapor. Cesium has been chosen as the working medium since the frequency change effect of the center of the magnetic resonance line with the field sign change, inherent for alkali metals, has the lowest value for cesium and is about 1–2 nT for this magnetometer. The magnetometer design provided measurements with an arbitrary orientation of the spacecraft. To avoid any possible dead areas the magnetometer had two absorption chambers oriented at 135° angle to each other. The instrumental accuracy of the magnetometer was estimated at the level of 1.7 nT. The correspondence of the quantum-cesium magnetometer readings to the absolute values in the range of measured fields was verified by comparison with the proton magnetometer. Correspondence was within 2 nT. The magnetometer sensors were placed in a special unpressurized container mounted on a deployable boom of 3.6 m length. The magnetometer was designed and manufactured by IZMIRAN and its experimental design bureau (Dolginov et al., 1970). The external view of the DS-U2-MG type spacecraft is given in Fig. 2.

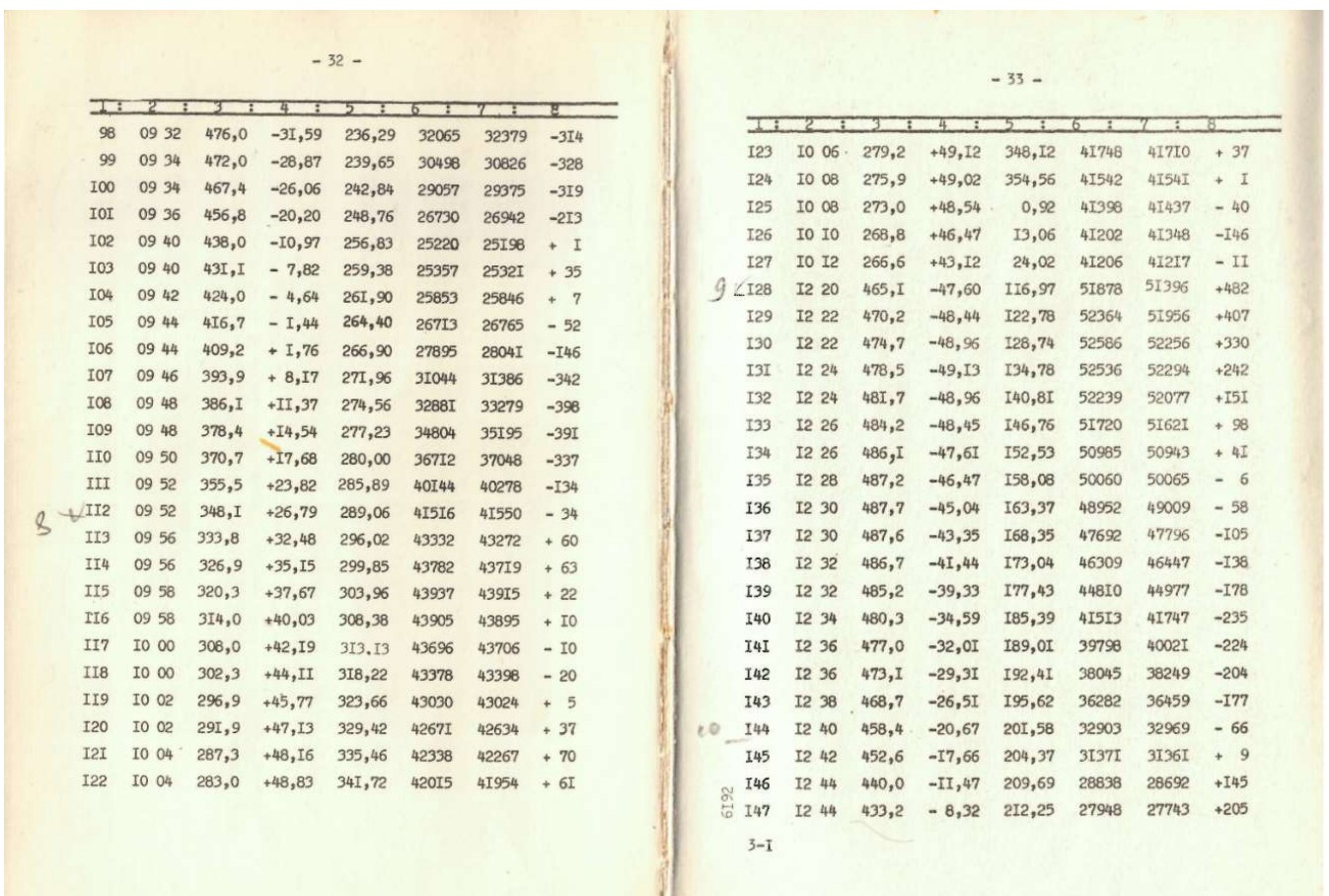

**Figure 3: A sample page of Kosmos-49 catalogue (Dolginov et al., 1967).**

It should be mentioned that at the same period of 1965-1971 the satellites of POGO series flew, in particular OGO-2, OGO-4
and OGO-6, those magnetic data are available from NASA (Langel and Hinze, 1998).

## 2    Data description

Geomagnetic measurements by Kosmos-49 satellite were carried out in the framework of the international program of the
World Magnetic Survey (Benkova and Dolginov, 1971). The Kosmos-49 mission objectives were as follows: global survey
of the Earth's magnetic field and compilation of a map of its spatial distribution; refinement of Gaussian coefficients of
magnetic potential decomposition; investigation of the Earth's magnetic field secular variation and temporal changes at the
altitudes of the spacecraft flight in the magnetoactive periods. Kosmos-49 operated from October 24 to November 3, 1964,
totally 11 days. It performed 162 orbits around the Earth and made 17300 measurements covering 75% of the Earth's surface
almost homogeneously. It had an orbit with inclination of 49°, nodal period of 91.83 min, apogee of 484 km and perigee of
265 km. During the mission the apogee decreased from 487 to 472 km approximately linearly in time (see Fig. 4). The
accuracy of satellite position was 3 km along the trajectory and 1 km in transverse direction, the accuracy of timing was
about 0.5 s.

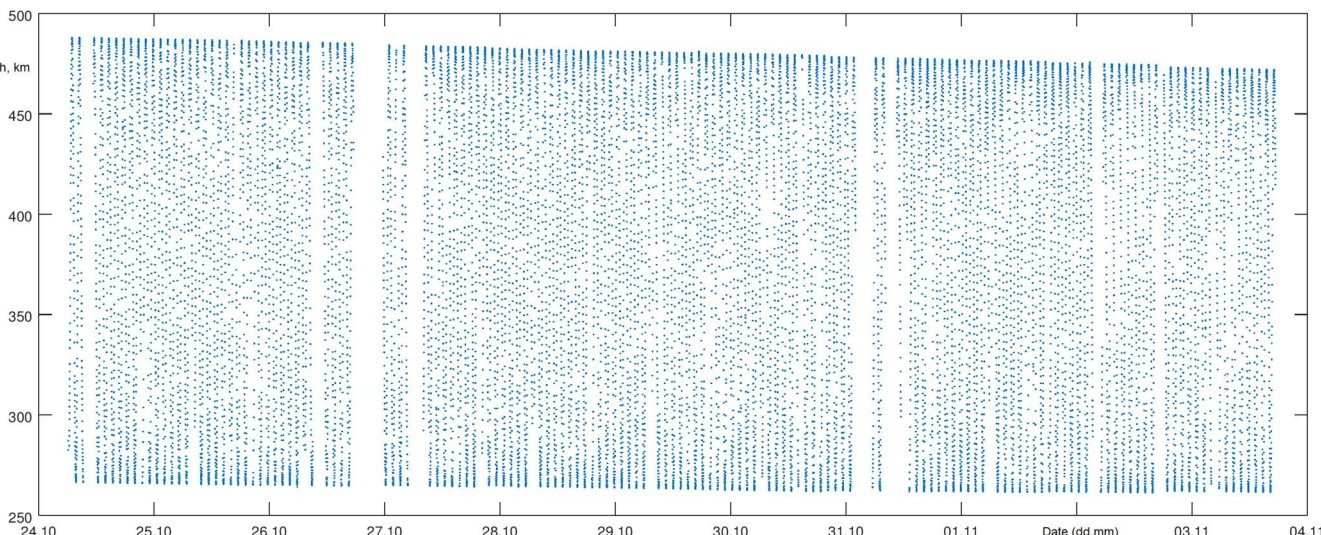

**Figure 4: The depenedence of Kosmos-49 altitude on time.**

Homogenous survey, performed within a short period of time, provided the general image of the Earth's magnetic field free
of secular variations and allowed to map its distribution on the date of the experiment. The collected data were used for
obtaining the international analytical model of the Earth's magnetic field. These results were presented and obtained a wide

scientific recognition at the 7th General Assembly of the ICSU Committee on Space Research (COSPAR) in Florence (Italy) in 1964 (Konyukhov, 2000).

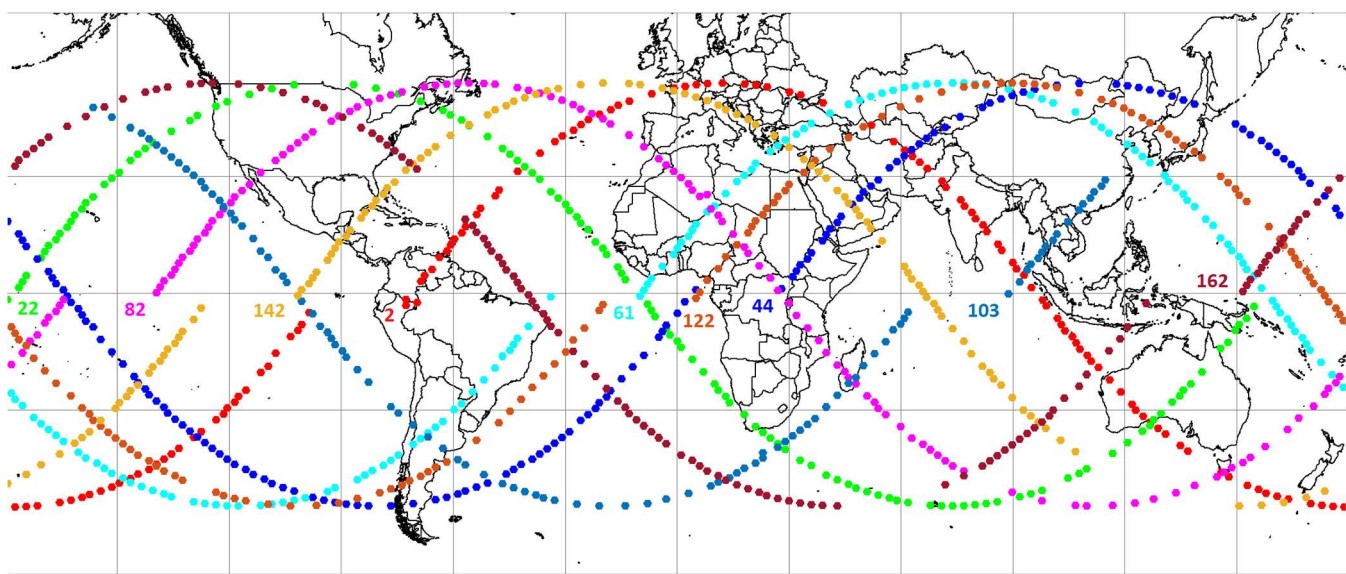

**Figure 5: Spatial coverage of Kosmos-49 data. The projection of approximately every 20th orbit on the Earth's surface is shown. Dots indicate the locations of measurements.**

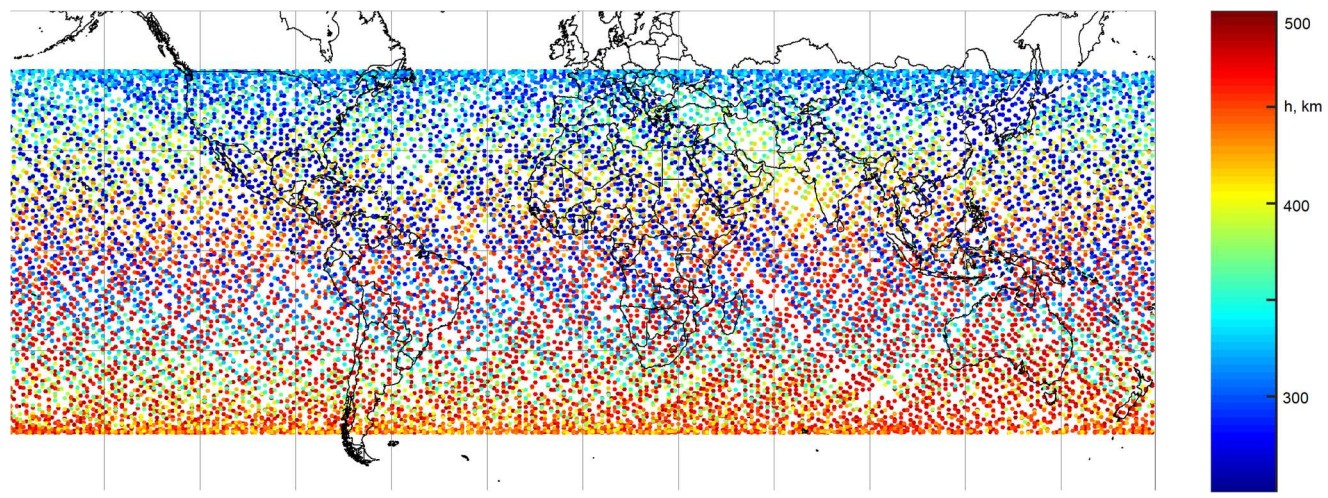

**Figure 6: Kosmos-49 positions of measurements. Altitude is shown in color.**

The sample page of the catalogue (Dolginov et al., 1967) can be seen in Fig. 3. It contains 8 columns with the following data: 1) number of measurement; 2) Moscow time (hours and minutes, rounded to 2 min); 3) altitude, km; 4) latitude (north positive), degrees; 5) longitude (positive, from Greenwich eastward), degrees; 6) measured magnetic field absolute value,

nT; 7) magnetic field value calculated according to global model presented in (Adam, 1964), nT; 8) the difference between measured and calculated values, nT.

The evolution of Kosmos-49 altitude is shown in Fig. 4. The spatial coverage of data is shown in Fig. 5. The projection of approximately every 20th orbit on the Earth's surface is shown. Dots indicate the locations of measurements. The gaps correspond to areas in which the data were not recorded due to technical failures.

In Figures 6 and 7 the complete set of satellite positions at the moments of measurements and the map of measured magnetic field absolute value are shown.


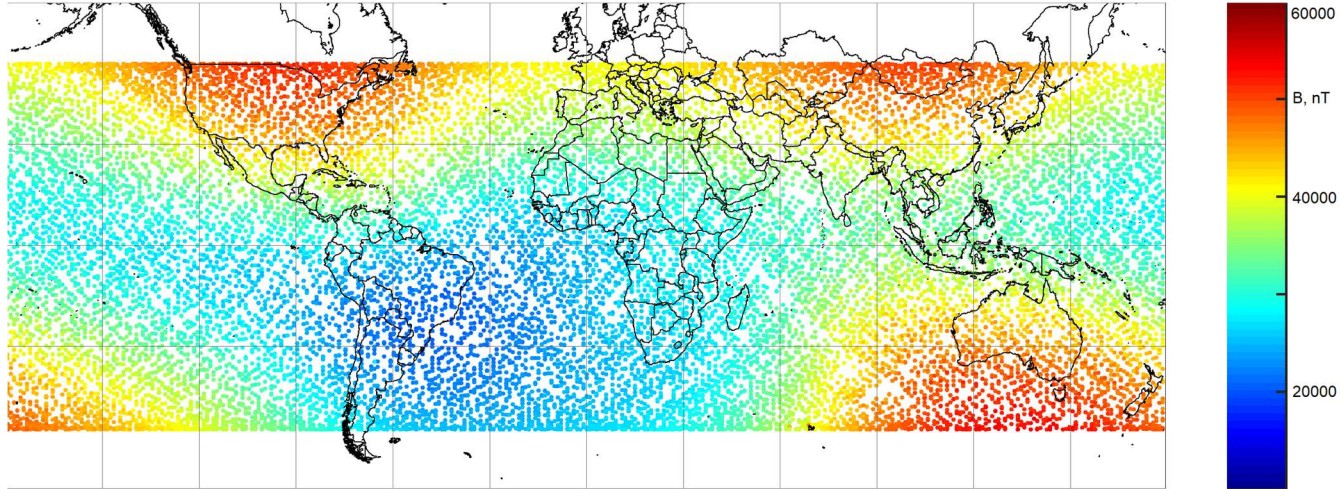

**Figure 7: Magnetic field map measured by Kosmos-49.**

The Kosmos-321 mission main objectives were the same as for Kosmos-49, but also included some specific tasks, such as 125    confirming the possibility of using magnetometers with optical pumping as a service system for nuclear test explosion control in outer space and a series of atmospheric experiments. The mission experimental program was developed by IZMIRAN and Space Research Institute of the Academy of Sciences of the USSR. Kosmos-321 operated from January 20 to March 13, 1970, totally 52 days. It performed 823 orbits around the Earth and made over 600000 measurements covering 94% of the Earth's surface. It had an orbit with inclination of 71°, nodal period of 92 min, apogee of 507 km and perigee of 130    280 km. During the mission the apogee decreased from approximately 500 to approximately 300 km. In the available data the apogee runs from 470 to 315 km.

Measurements were performed with 2 s sampling. The primary examination of data showed that it contains strong interference (up to 20 nT) presumably from thermocurrents in sensor fixing device. (The effect of thermocurrents was

reproduced on the fixing device similar to that of Kosmos-321.) This interference exhibits itself as a gap in measured
magnetic field absolute values every time the sensor used for measurement was changed (see Fig. 8). For this reason the
catalog contains data for only limited number of orbits, a total of 5000 measurements, for which the interference has
approximately sinusoidal form and was eliminated from data upon data processing at IZMIRAN. The "equal squares"
method was applied. Both raw data and corrected data are presented in the catalogue (Dolginov et al., 1976). It should be
noted that catalog uses 20 s sampling, not 2 s sampling used during actual measurements.


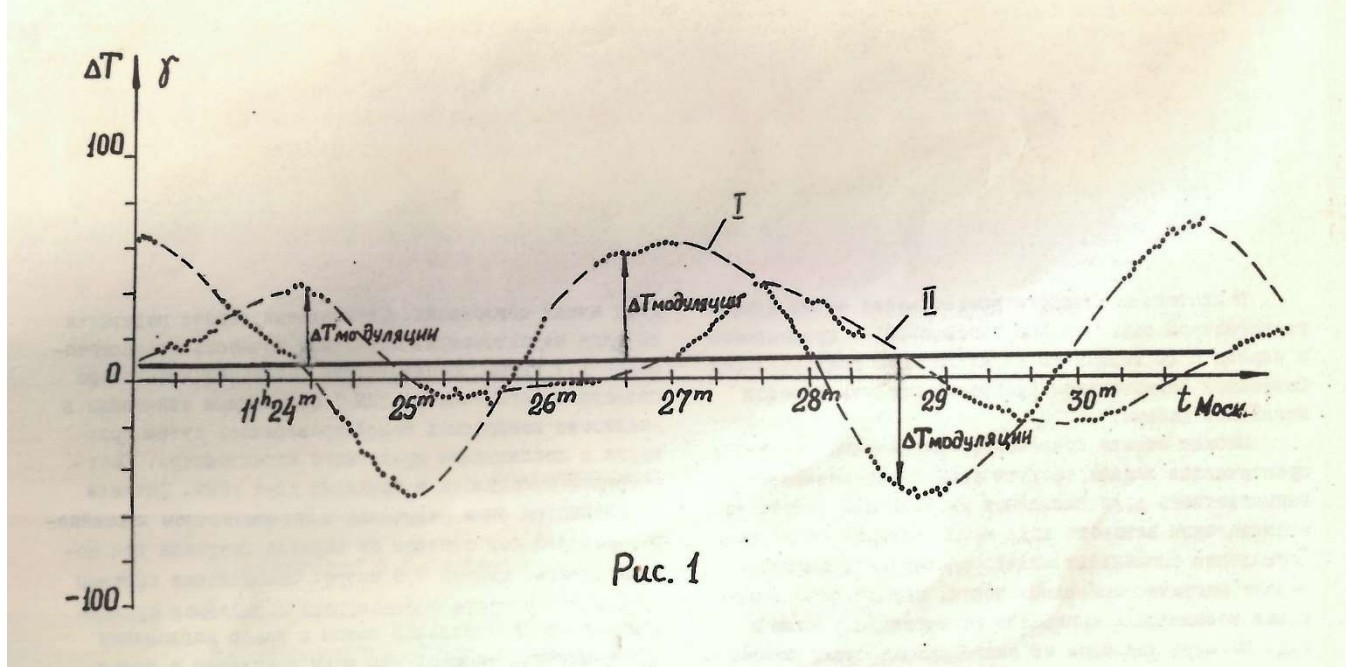

**Figure 8: Interference in Kosmos-321 data as shown in the catalog (Dolginov et al, 1976). Roman I and II stand for two sensors, actual measurements are indicated with dots, dashed lines continuously complement the measurements of each sensor.**

The sample page of the catalogue can be seen in Fig. 9. It contains 9 columns with the following data: 1) Moscow time
(hours, minutes, and seconds); 2) local time; 3) latitude (north positive), degrees; 4) longitude (positive, from Greenwich
eastward), degrees; 5) altitude, km; 6) measured magnetic field absolute value, nT; 7) corrected magnetic field absolute
value, nT; 8) the difference between corrected value and the value calculated according to global model presented in (Cain,
1970), nT; 9) calculated magnetic inclination, degrees.

The dependence of Kosmos-321 altitude on time is shown in Fig. 10. Spatial coverage of Kosmos-321 data is shown in Fig.
11 and corrected magnetic field value is shown in Fig. 12.

The Kosmos-49 mission data tables occupied 648 pages (three volumes) (Dolginov et al., 1967), and the Kosmos-321 mission data tables occupied 173 pages (Dolginov et al., 1976). The data were digitized by the Laboratory of Geophysical Data of Geophysical Center of RAS within the three summer months of 2019. For the convenience of users, in the digital version of the catalogues the table structure has been slightly changed as follows.

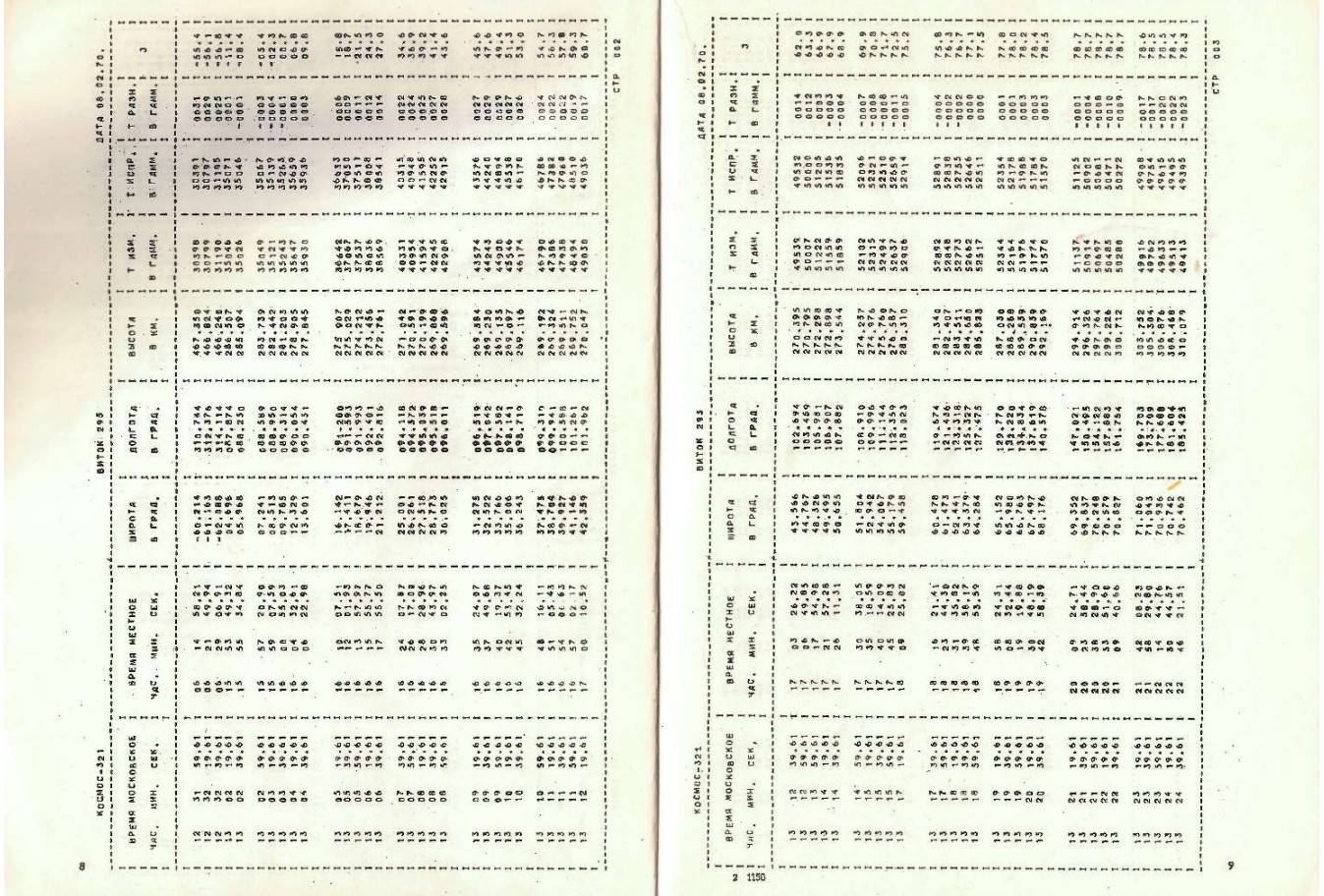

**Figure 9: A sample page of Kosmos-321 catalogue (Dolginov et al., 1976).**

Kosmos-49 columns:

| Julian day | Date/Time (Moscow) yyyy-mm-ddThh:mm | Number of measurement | Altitude, km | Latitude, deg | Longitude, deg |
| --- | --- | --- | --- | --- | --- |
| | | | | | |

| T measured, nT | T calculated, nT | $\Delta T = T_{meas} - T_{calc}$, nT | Orbit number |
| --- | --- | --- | --- |
| | | | |

Kosmos-321 columns:

| Julian day | Date/Time (Moscow) yyyy-mm-ddThh:mm:ss.fff | Date/Time (local) yyyy-mm-ddThh:mm:ss.fff | Orbit number |
|---|---|---|---|
| | | | |

| Latitude, deg | Longitude, deg | Altitude, km | T measured, nT | T corrected, nT | $\Delta T = T_{corr} - T_{calc}$, nT | Calculated inclination, deg |
|---|---|---|---|---|---|---|
| | | | | | | |

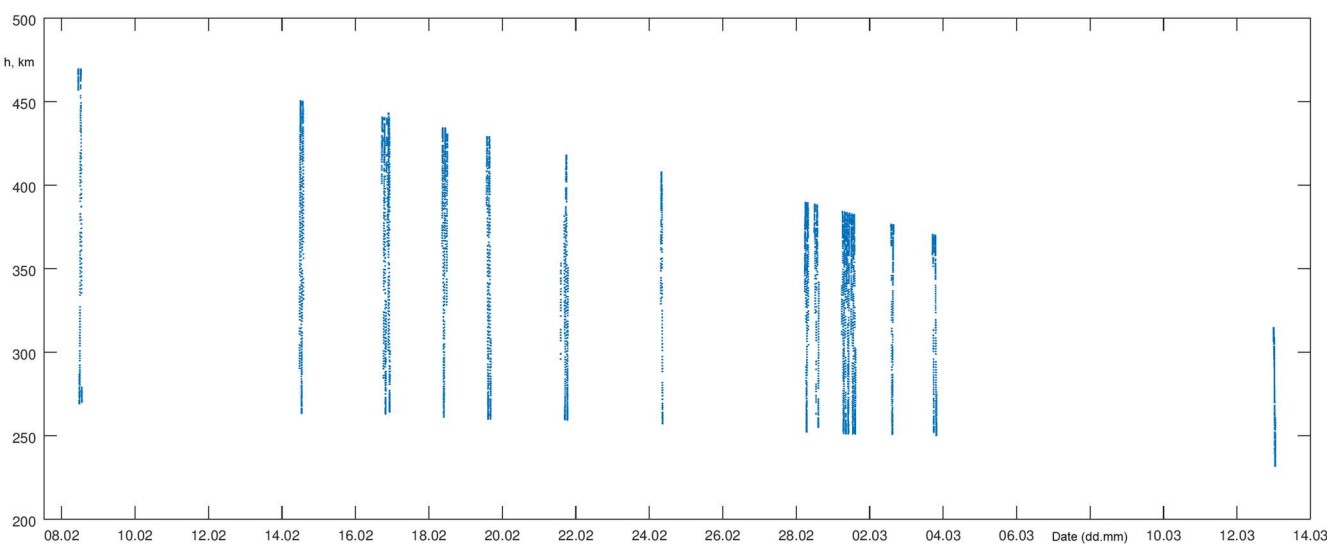

**Figure 10: The dependence of Kosmos-321 altitude on time.**

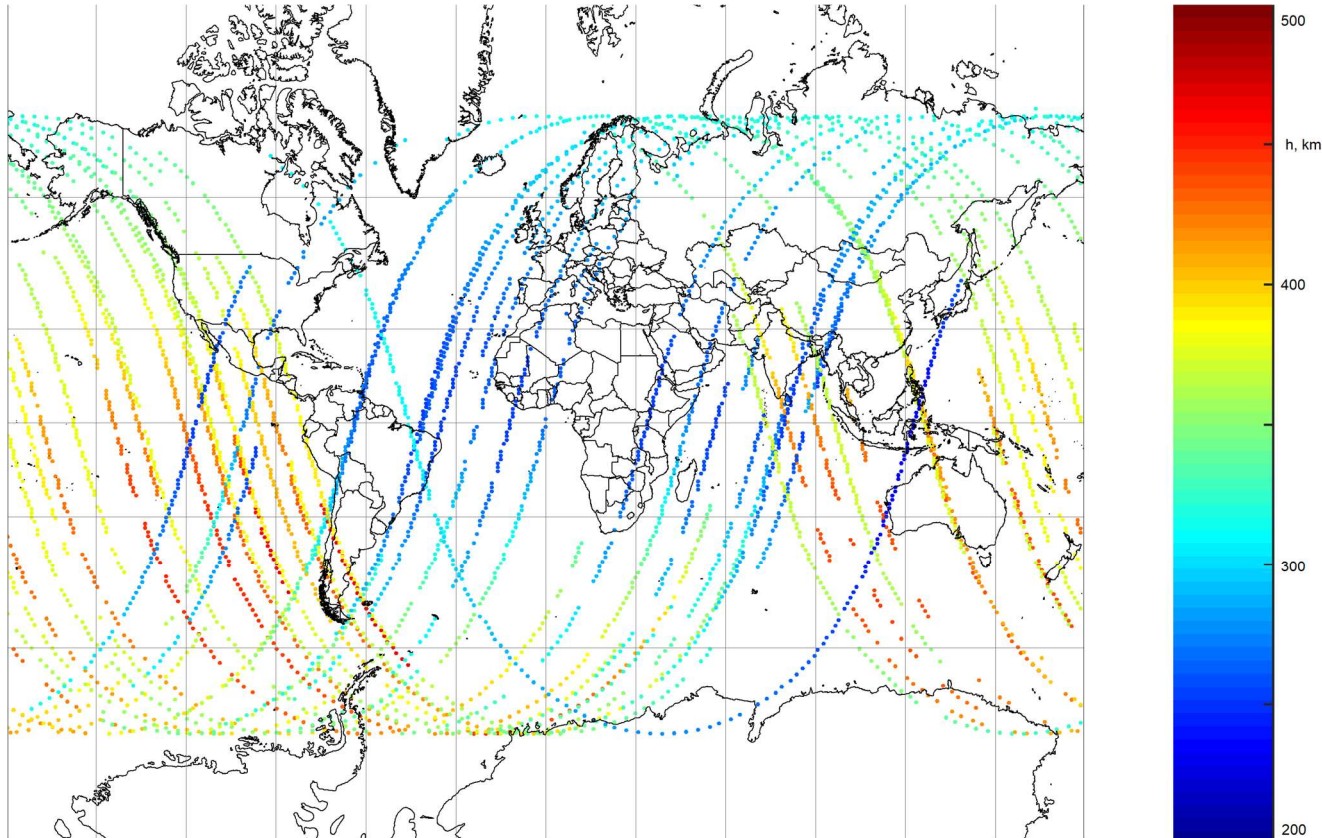

**Figure 11: Spatial coverage of Kosmos-321 available data.**


It is obvious that digitization of the data set of such a volume could have led to errors and misprints made by the digitizing personnel. To prevent such a situation all the data arrays were automatically checked after the digitization. It was done by means of specially developed software that for each orbit revolution of the spacecraft controlled the time intervals, allowed to plot the graphs for the spacecraft orbital motion and for the measured and calculated geomagnetic data. The consistency,

monotony, smoothness, extremal points, etc. of the digitized data were checked. In case of discrepancies within the software calculation results, manual check with the printed versions of catalogues was carried out. These operations minimized all possible errors of the personnel during digitization. The digitized catalogues are publicly available in ASCII tab separated text format (Krasnoperov et al., 2020).


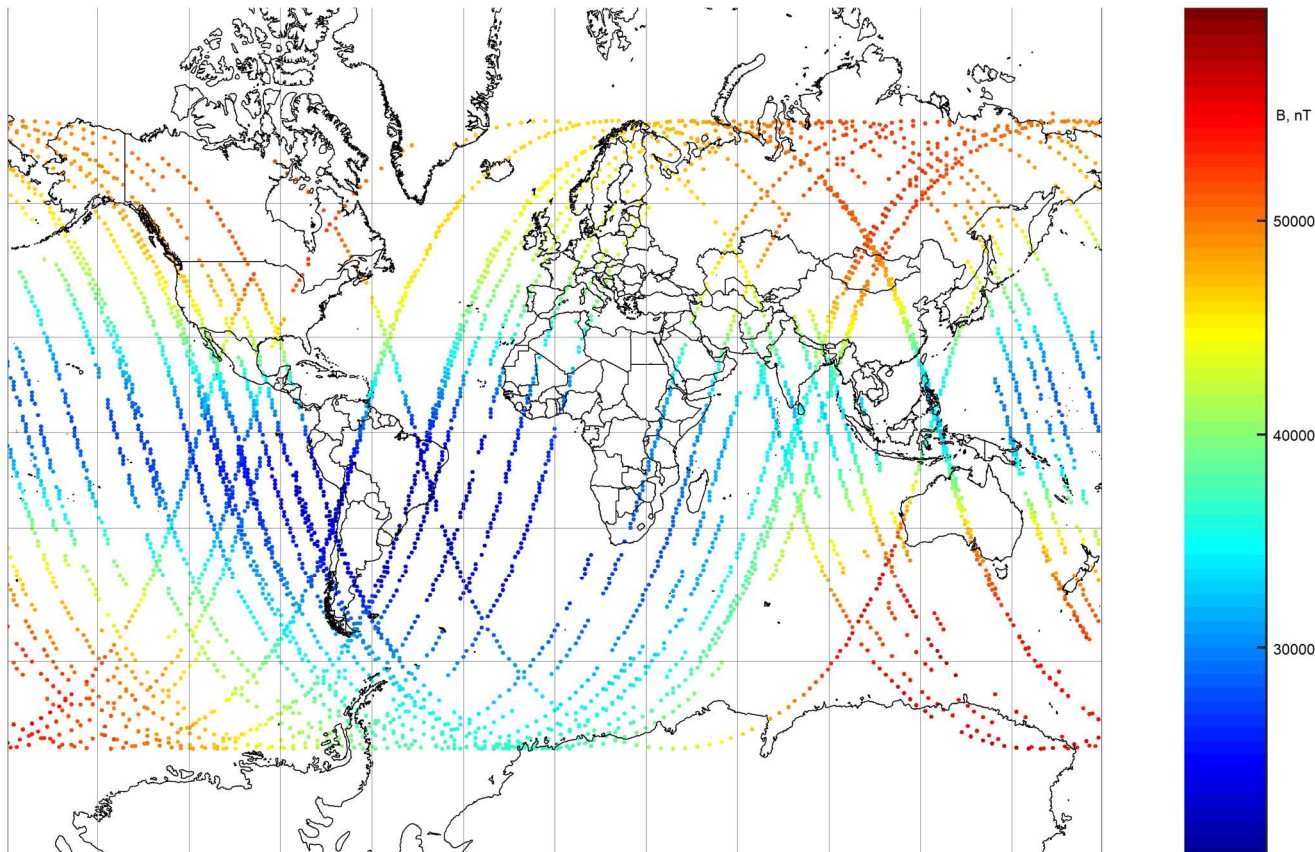

**Figure 12: Kosmos-321 measured magnetic field.**

## 4 Conclusions

The success of satellite missions of Kosmos-49 and Kosmos-321 spacecraft allowed unique data on the spatial distribution of the total intensity of the Earth's magnetic field over almost the whole surface of the planet to be obtained. This data being the part of the international program of the World Magnetic Survey were submitted to the main World Data Centers in Moscow (USSR), Maryland (USA), Charlottenlund (Denmark), Kyoto (Japan) and were used as initial data for analysis of the structure of the Earth's magnetic field sources and for compilation of a series of its analytical models. The most notable model that employed Kosmos-49 data was the first generation of the International Geomagnetic Reference Field (for epoch 1965.0) (Zmuda, 1971). The model was agreed by a working group on October 24, 1968 in Washington, D.C. (USA), approved by a special committee of the International Association of Geomagnetism and Aeronomy in February 1969 and

officially introduced at the XVth General Assembly of the International Union of Geodesy and Geophysics in 1971 in
Moscow (Blagonravov, 1978).

Comparison of the data collected by Kosmos-49 and Kosmos-321 spacecraft allowed the determination of the Earth's magnetic field secular variation with high accuracy for the period of 1965–1970. Another important topic was the study of temporal changes in the geomagnetic field at the altitudes of the flight of spacecraft during magnetoactive periods. Kosmos-321 encountered a strong magnetic storm of March 8–10, 1970. As a result, very interesting and important data on the
magnetic storm mechanisms in polar regions were obtained. However, the data registered during this period was not included in publicly available catalogue. Only a few footprints of these data may be found in paper (Dolginov et al., 1972). Kosmos-321 for the first time measured the effects of the equatorial electrojet (Vanian et al., 1975). In addition to the traditional known magnetic storm current systems detected by ground based magnetic observatory records, magnetic effects of currents along power lines that were not detected by the nearest magnetic observatories were also revealed.

The results of the Kosmos-49 and Kosmos-321 missions became the sound base for further fundamental studies of the Earth's magnetic field. The uniqueness of the presented data is emphasized by the fact that there are practically no older and publicly available global satellite data on the Earth's magnetic field.

**6 Data availability**

The data from the paper catalogues were digitized at Geophysical Center of RAS in 2019. Digital data are available at
PANGAEA (Krasnoperov et al., 2020, https://doi.org/10.1594/PANGAEA.907927).

**Author contributions.** RK, RL and AS – preparation of the manuscript. DP – description and preliminary analysis of initial data, publication of the digitized catalogues. BD – data digitalization management, data validation, description of digitized data, and preparation of figures.

**Competing interests.** The authors declare that they have no conflict of interest.

**Acknowledgements.** The authors wish to thank the Institute of Terrestrial Magnetism, Ionosphere and Radio Wave Propagation of AS USSR and the Space Research Institute of AS USSR for compilation of printed versions of the presented catalogues. This work employed data and services provided by the Shared Research Facility "Analytical Geomagnetic Data Center" of the Geophysical Center of RAS (http://ckp.gcras.ru/). The authors wish to thank the team of the World Data Center for Solar-Terrestrial Physics and World Data Center for Solid Earth Physics in Moscow (Russia) for preservation and
making publicly available historical geophysical recordings.

**Financial Support.** This work was conducted in the framework of budgetary funding of the Geophysical Center of RAS, adopted by the Ministry of Science and Higher Education of the Russian Federation.

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
