# Peer review of "Early Soviet satellite magnetic field measurements from 1964 and 1970"

_Earth System Science Data, 2019_

## Referee Comment (RC1) · Anonymous Referee #1 · 15 Dec 2019

The submitted paper describes the Earth's magnetic field satellite data from two early Russian satellite mission, Kosmos-321 and Kosmos-49. These data have are archived on microfiche and tables, and their digitization and publication are a valuable and important contribution to the accessibility of satellite magnetic field data. Therefore, I strongly encourage publication of this data set in digital form.

The paper describes the main aspects of the data and the underlying satellite mission. Still, the description of the dataset and its format could profit from more details and some more figures. Particularly, I have the following suggestions to make:

- Include scatter plots similar to Fig. 4, but for all data, and showing field magnitude and orbit altitude using colors, for each of the two missions (four plots in total)

[Figure]

- Show a plot of orbit altitude vs. time, in order to better understand the altitude evolution of the satellites

- P.4, L85: Add information on decaying apogee with time during the mission (decaying from 487km to 472 km)

- The description of the dataset columns should be visualized as table (see p. 7, L127, and following lines)

- Are data of Kosmos-26 and Kosmos-356 missions also available somewhere?

- P.5, L95: If available, some information on occurrence rate and type of technical failures would be interesting

- P.6 , L108: 600000 measurements: Is this the total number of data created including all type of instruments? Where does this number come from?

- P.6, L108: "94% coverage": Plotting the data in a lon-lat-coordinate system, longitudinal data gaps are obvious. How did you derive this number - i.e. what area bin size did you use to define spatial data coverage?

- P.6, L109: Also here, please mention evolution of apogee, decaying from ∼500 to ∼300 km

- P.6, L110: In the data, 20s sampling rate is reported. Was the data decimated or is it a typo?

- P.6, L114: Is there more information on the applied correction available?

Concerning the dataset itself:

- Time should also be included in a more convenient format, e.g. JD2000 / UTC

- Columns in datasets for Kosmos-321 and Kosmis-49 mission should be consistent

- The 'Device' column is unnecessary as the dataset is split anyway

- The 'Orbit number' column is missing for some data

- The header of Kosmos-49 data says '55162 data points', but there are only 8888 data points in the table. Similar error exists for Kosmos-321 data.

- The data time range on the DOI landing page is wrong for Kosmos-321 (whole year 1970)

- Can the orbits / data points be better visualized on PANGEA landing page?

Technical comments:

- Dolginov, 1965 is missing in the bibliography

---

## Referee Comment (RC2) · Susan Macmillan (Referee) · 20 Dec 2019

This paper presents digitised data from Soviet satellite magnetic surveys in 1964 and 1970 and provides some background information to accompany them. The data represent some of the earliest space-based magnetic field measurements around and their publication is to be welcomed. How do you estimate 75% and 94% coverage in the abstract? The section "Satellite missions for magnetic field measurements" is hard to follow, partly because of more than one name for each satellite, and would benefit from splitting into 1.1 Kosmos-49 and 1.2 Kosmos-321. The paper would benefit from information on how the satellite positions were determined and how the timing was achieved, and if possible, a comment on their accuracy. If anything is known about what was done to estimate spacecraft fields at the magnetometer sensors this should

also be included. These factors will have a huge influence on the accuracy of the final magnetic data. If you know the dimensions of Kosmos-49, provide it. It would be good to have, as far as possible, equivalent information for both satellites. It would be interesting to have the authors' opinion on the accuracy for Kosmos-49 quoted by Benkova and Dolginov (1971) – do they agree with 25-30 nT? Can an equivalent estimate be made of the accuracy of the data from Kosmos-321? In Figure 2 clarify where the 2 absorption chambers are? We are told they are at 135° to one another but the boom appears to be at 135° to satellite body. Figure 4 – this does not show the complete coverage of Kosmos-49 data. Clarify that it is only showing a certain number of (near-complete?) orbits. I plotted the whole dataset and the orbits disappear for Kosmos-49 but there is good even coverage. Include plots in the paper showing complete coverage of both satellites, and for added interest and confidence in the data, the data positions can be colour-coded according to magnetic field measured. The area of weak field, called the South Atlantic Anomaly, shows up nicely, as well as larger magnetic fields at high latitudes. Examples included. The altitude of the digitised Kosmos-321 data is 232-470 km. This is somewhat different to the 280-507 km range given in the paper. Please check. The POGO series of satellites should be mentioned, in particular OGO-2, OGO-4 and OGO-6 which flew 1965-1971, and whose magnetic data are readily available from NASA. A suitable reference would be Langel and Hinze, 1998. "The magnetic field of the Earth's lithosphere: the satellite perspective". Detailed corrections Title - should be "Early Soviet satellite magnetic field measurements from 1964 and 1970". Abstract - replace "Totally" in abstract with "A total of" Line 21 - replace "to analyse of" with "the analysis of" Line 32 – remove "totally" Line 59 – replace "to organize" with "the organisation of" Line 159 – remove "to obtain" and add "to be obtained" at end of sentence Line 169 – replace "to determine" with "the determination of"

Please also note the supplement to this comment:
https://www.earth-syst-sci-data-discuss.net/essd-2019-218/essd-2019-218-RC2-supplement.pdf

[Figure]

**Fig. 1.** Kosmos-49_1

**Fig. 2.** Kosmos-49_2

**Fig. 3.** Kosmos-321

---

## Author Comment (AC1) · 16 Jan 2020

Below are the answers on referees' comments. The referees' comments are indicated with R1 and R2, our answers are indicated with A. Revised version of the paper is attached. All appropriate changes in the revised version are marked with green for convenience.

R1: Include scatter plots similar to Fig. 4, but for all data, and showing field magnitude and orbit altitude using colors, for each of the two missions (four plots in total)

R2: Include plots in the paper showing complete coverage of both satellites, and for added interest and confidence in the data, the data positions can be colour-coded according to magnetic field measured.

[Figure]

A: Done. See Figures 6, 7, 11, 12 at lines 110, 124, 173, 188 and text insertions at lines 119 and 151.

R1: Show a plot of orbit altitude vs. time, in order to better understand the altitude evolution of the satellites

A: Done, see Figures 4 and 10 at lines 99 and 159 and text insertions at lines 116 and 151.

R1: P.4, L85: Add information on decaying apogee with time during the mission (decaying from 487 km to 472 km)

A: Done, see line 95.

R1: The description of the dataset columns should be visualized as table (see p. 7, L127,and following lines)

A: Done, see lines 160-164.

R1: Are data of Kosmos-26 and Kosmos-356 missions also available somewhere?

A: No, to our knowledge.

R1: P.5, L95: If available, some information on occurrence rate and type of technical failures would be interesting

A: Unfortunately, we don't have any information on these topics.

R1: P.6 , L108: 600000 measurements: Is this the total number of data created including all type of instruments? Where does this number come from?

P.6, L110: In the data, 20s sampling rate is reported. Was the data decimated or is it a typo?

A: It is the number of magnetic field absolute value measurements. No different measurements were performed by satellite. The number comes from the description of catalog. One can also make elementary estimates. The lifetime of satellite was 52

days, the measurements were performed 10 hours per day, the sampling rate was 2 s, thus possible number of measurements was approximately 900000. Due to technical failures real number of measurements is less.

Sampling rate of 2 s is not a typo. If it were 20 s, the number of measurements would not excess 90000 contrary to reported 600000. Also the new Figure 8, illustrating the interference, definitely says the sampling rate was 2 s not 20 s. I have no idea why 20 s sampling rate was chosen for catalog data. I added the note on difference between actual sampling rate and that of the data, see line 140.

R1: P.6, L108: "94% coverage": Plotting the data in a lon-lat-coordinate system, longitudinal data gaps are obvious. How did you derive this number - i.e. what area bin size did you use to define spatial data coverage?

R2: How do you estimate 75% and 94% coverage in the abstract?

A: It is estimated with the maximal latitude value. If minimal co-latitude equals $\theta$ then the square of two uncovered polar caps equals $2 \cdot 4\pi \sin^2(\theta/2)$, while total spherical square is $4\pi$. Maximal latitudes for Kosmos-49 and Kosmos-321 equal 49 and 71 degrees, these result in 75% and 94% coverage correspondingly.

R1: P.6, L109: Also here, please mention evolution of apogee, decaying from 500 to 300 km

A: Done, see line 131.

R1: P.6, L114: Is there more information on the applied correction available?

A: Very few technical details are presented in the description of the catalog. No references are made to more comprehensive description of the procedure applied. We reproduced the details found in the description of the catalog, see Figure 8 at line 141 and text insertions at lines 133-140.

R1: Columns in datasets for Kosmos-321 and Kosmos-49 mission should be consistent

A: Since the data presented is historical and subject to errors we would like to preserve the original data structure found in the catalogs. In our opinion the additions are allowed but the omissions are not. Since the original catalogs are inconsistent the data will be inconsistent anyway. The only thing we can do is to place additional columns in coherent manner.

R1: Time should also be included in a more convenient format, e.g. JD2000 / UTC

The 'Device' column is unnecessary as the dataset is split anyway

The 'Orbit number' column is missing for some data

The header of Kosmos-49 data says '55162 data points', but there are only 8888 data points in the table. Similar error exists for Kosmos-321 data.

The data time range on the DOI landing page is wrong for Kosmos-321 (whole year1970)

A: The request for corrections is sent to PANGAEA team.

R1: Can the orbits / data points be better visualized on PANGEA landing page?

A: I can't answer this question, it is the responsibility of PANGAEA team.

R1: Dolginov, 1965 is missing in the bibliography

A: It's a typo in the text, Dolginov, 1966 should stay instead. Corrected, see line 47.

R2: The section "Satellite missions for magnetic field measurements" is hard to follow, partly because of more than one name for each satellite, and would benefit from splitting into 1.1 Kosmos-49 and 1.2 Kosmos-321.

A: Done.

R2: The paper would benefit from information on how the satellite positions were determined and how the timing was achieved, and if possible, a comment on their accuracy.

A: The satellites were launched and operated by Soviet Ministry of Defence. Nothing is known how satellite positions were determined. The timing was achieved by on-board clock. The Kosmos-49 catalog says that the accuracy of satellite position was 3 km along the trajectory and 1 km in transverse direction, the accuracy of timing was about 0.5 s. I added this information, see lines 96. No information on accuracy of position and time for Kosmos-321 is available.

R2: If anything is known about what was done to estimate spacecraft fields at the magnetometer sensors this should also be included.

A: As for Kosmos-49 satellite, the magnetic effects of its body were compensated to accuracy of 2 nT by the system of permanent magnets, I have edited the text, see line 53. No information is available on Kosmos-321. It is known that Kosmos-321 data contains strong interference (up to 20 nT) presumably from thermocurrents in sensor fixing device, see lines 133-140 .

R2: If you know the dimensions of Kosmos-49, provide it.

A: Done, see line 52.

R2: It would be interesting to have the authors' opinion on the accuracy for Kosmos-49 quoted by Benkova and Dolginov (1971) – do they agree with 25-30 nT? Can an equivalent estimate be made of the accuracy of the data from Kosmos-321?

A: The analysis of Benkova and Dolginov is based on a comparison of the magnetic field values measured on n-th and n+77-th orbits, which are approximately coincident. We find it the most direct and reliable method. The equivalent estimate for Kosmos-321 is impossible, since we don't have sufficient number of coincident orbits.

R2: In Figure 2 clarify where the 2 absorption chambers are? We are told they are at $135°$ to one another but the boom appears to be at $135°$ to satellite body.

A: Both chambers are situated in the red cylinder at the end of the boom. The angle between the boom and the satellite axis has nothing to do with the angle between the

axes of the chambers.

R2: Figure 4 – this does not show the complete coverage of Kosmos-49 data. Clarify that it is only showing a certain number of (near-complete?) orbits.

A: Approximately every 20th orbit is shown (the figures on the plot are actual orbit numbers), it is mentioned in the text. I have doubled this information in the figure caption, see line 106.

R2: The altitude of the digitized Kosmos-321 data is 232-470 km. This is somewhat different to the 280-507 km range given in the paper. Please check.

A: There is no contradiction. The catalog presented contains only a small portion of complete data over reduced time interval (8.02-13.03 versus 20.01-13.03). The range 280-507 km is the initial altitude range.

R2: The POGO series of satellites should be mentioned, in particular OGO-2, OGO-4 and OGO-6 which flew 1965-1971, and whose magnetic data are readily available from NASA. A suitable reference would be Langel and Hinze,1998. "The magnetic field of the Earth's lithosphere: the satellite perspective".

A: Done, see line 85 and reference list.

R2: Title - should be "Early Soviet satellite magnetic field measurements from1964 and 1970".

Abstract - replace "Totally" in abstract with "A total of"

Line 21 - replace "to analyse of" with "the analysis of"

Line 32 – remove "totally"

Line 59 – replace "to organize" with "the organisation of"

Line 159 – remove "to obtain" and add "to be obtained" at end of sentence

Line 169 – replace "to determine" with "the determination of"

A: Done, see lines 1, 9, 22, 34, 64, 191, 200.

Please also note the supplement to this comment:
https://www.earth-syst-sci-data-discuss.net/essd-2019-218/essd-2019-218-AC1-supplement.pdf

**Supplement:**

[revised manuscript text omitted]

---

## Editor Decision (ED1)

Dear Dimitry Krasnoperov et al,

Many thanks for the careful revision of your manuscript that is ready for publication after some minor technical corrections.

1. Please contact PANGAEA and ask them to register the DOI. The current doi.pangaea.de link is only preliminary and not accepted for the final version of ESSD papers.
2. Please change the DOI link in the manuscript from https://**doi.pangaea.de**/10.1594/PANGAEA.907927 to https://**doi.org**/10.1594/PANGAEA.907927 throughout the manuscript (abstract; data availability statement, references). Please make sure to note the correct publication year, it is currently 2019, but I assume that the final version
3. Page 11, Line 179-183: it is not necessary to include the full citation of the data in the body text. Please change the sentence to "The digitized catalogues are publicly available in ASCII tab separated 180 text format (**Lukianova et al., 2020)**."

I have also found some bugs in the summary file of the data sets:

a) The citation is still referring to the "data in review" preliminary version (see point 1 above). Please update it with the final version of
b) Citation of child datasets (DOIs (/* TABULAR SUMMARY OF DATASETS LISTED IN THIS COLLECTION: */) is repeating the parent citation with DOI (duplications).

According to the landing page on Pangaea, the child datasets should be cited as:

Lukianova, R; Peregoudov, D; Dzeboev, B et al. (2019): Magnetic field measurements from the satellite Kosmos-321. https://**doi.org**/10.1594/PANGAEA.907926

Lukianova, R; Peregoudov, D; Dzeboev, B et al. (2019): Magnetic field measurements from the satellite Kosmos-49, measured with proton magnetometer 1. https://**doi.org**/10.1594/PANGAEA.907923

Lukianova, R; Peregoudov, D; Dzeboev, B et al. (2019): Magnetic field measurements from the satellite Kosmos-49, measured with proton magnetometer 2. https://**doi.org**/10.1594/PANGAEA.907924

I don't know if this summary is automatically created by PANGAEA or if you composed it. If it was provided by PANGAEA, please feel free to forward this information to the PANGAEA team.

Please send the revised version of the manuscript via email to anna.wenzel@copernicus.org

Many thanks for choosing ESSD

Best regards,

Kirsten Elger

(ESSD Chief Executive Editor)

---

## Author Response (AR2)

**Answers on editor's comments.**

Editor comments file seems a little corrupted, some line endings are dropped, I reproduce it "as is".
* * *
**E:** Please contact PANGAEA and ask them to register the DOI. The current doi.pangaea.de link is only preliminary and not accepted for the final version of ESSD papers.

The citation is still referring to the "data in review" preliminary version (see point 1 above). Please update it with the final version of

**A:** Done. DOI is registered, "data in review" comment is removed automatically.
* * *
**E:** Please change the DOI link in the manuscript from https://**doi.pangaea.de**/10.1594/PANGAEA.907927 to https://**doi.org**/10.1594/PANGAEA.907927 throughout the manuscript (abstract; data availability statement, references). Please make sure to note the correct publication year, it is currently 2019, but I assume that the final version

**A:** Done, the DOI link is corrected throughout the manuscript, see lines 10, 208, 249. The final publication year is 2020.
* * *
**E:** Page 11, Line 179-183: it is not necessary to include the full citation of the data in the body text. Please change the sentence to "The digitized catalogues are publicly available in ASCII tab separated 180 text format (**Lukianova et al., 2020)**."

**A:** Done, see line 176. The author's order is changed (see **additional changes** below) so the citation now is (Krasnoperov et al., 2020).
* * *
**E:** Citation of child datasets (DOIs (/* TABULAR SUMMARY OF DATASETS LISTED IN THIS COLLECTION: */) is repeating the parent citation with DOI (duplications). According to the landing page on Pangaea, the child datasets should be cited as:

Lukianova, R; Peregoudov, D; Dzeboev, B et al. (2019): Magnetic field measurements from the satellite Kosmos-321. https://**doi.org**/10.1594/PANGAEA.907926
Lukianova, R; Peregoudov, D; Dzeboev, B et al. (2019): Magnetic field measurements from the satellite Kosmos-49, measured with proton magnetometer 1. https://**doi.org**/10.1594/PANGAEA.907923
Lukianova, R; Peregoudov, D; Dzeboev, B et al. (2019): Magnetic field measurements from the satellite Kosmos-49, measured with proton magnetometer 2. https://**doi.org**/10.1594/PANGAEA.907924

I don't know if this summary is automatically created by PANGAEA or if you composed it. If it was provided by PANGAEA, please feel free to forward this information to the PANGAEA team.

**A:** It found out to be a bug in PANGAEA software. The bug is now fixed and these lines are generated correctly. The PANGAEA teat expresses its gratitude for your valuable contribution to the improvement of their service.
* * *
**Additional changes.**

The second affiliation of Soloviev is omitted, see lines 3 and 4.

The authors order at PANGAEA is changed to match that in the article, see line 248 and citations in lines 10, 176, 208.

The description of datasets data columns is changed to match the final structure of datasets, see lines 157-161.

[revised manuscript text omitted]